# Genome Sequencing of *Pantoea agglomerans* C1 Provides Insights into Molecular and Genetic Mechanisms of Plant Growth-Promotion and Tolerance to Heavy Metals

**DOI:** 10.3390/microorganisms8020153

**Published:** 2020-01-22

**Authors:** Francesca Luziatelli, Anna Grazia Ficca, Mariateresa Cardarelli, Francesca Melini, Andrea Cavalieri, Maurizio Ruzzi

**Affiliations:** 1Department for Innovation in Biological, Agrofood and Forest systems (DIBAF), University of Tuscia, via C. de Lellis, snc, I-01100 Viterbo, Italy; f.luziatelli@unitus.it (F.L.); ficca@unitus.it (A.G.F.); 2CREA Research Centre for Vegetable and Ornamental Crops, I-84098 Pontecagnano, Italy; mteresa.cardarelli@crea.gov.it; 3CREA Research Centre for Food and Nutrition, Via Ardeatina 546, I-00178 Rome, Italy; francesca.melini@crea.gov.it; 4Department of Plant and Environmental Sciences, University of Copenhagen, DK–1871 Frederiksberg, Denmark; anca@plen.ku.dk

**Keywords:** *Pantoea agglomerans*, plant growth-promotion, *Solanum lycopersicum* L., indole-3-acetic acid, siderophores, arsenic resistance, complete genome, horizontal gene transfer

## Abstract

Distinctive strains of *Pantoea* are used as soil inoculants for their ability to promote plant growth. *Pantoea agglomerans* strain C1, previously isolated from the phyllosphere of lettuce, can produce indole-3-acetic acid (IAA), solubilize phosphate, and inhibit plant pathogens, such as *Erwinia amylovora*. In this paper, the complete genome sequence of strain C1 is reported. In addition, experimental evidence is provided on how the strain tolerates arsenate As (V) up to 100 mM, and on how secreted metabolites like IAA and siderophores act as biostimulants in tomato cuttings. The strain has a circular chromosome and two prophages for a total genome of 4,846,925-bp, with a DNA G+C content of 55.2%. Genes related to plant growth promotion and biocontrol activity, such as those associated with IAA and spermidine synthesis, solubilization of inorganic phosphate, acquisition of ferrous iron, and production of volatile organic compounds, siderophores and GABA, were found in the genome of strain C1. Genome analysis also provided better understanding of the mechanisms underlying strain resistance to multiple toxic heavy metals and transmission of these genes by horizontal gene transfer. Findings suggested that strain C1 exhibits high biotechnological potential as plant growth-promoting bacterium in heavy metal polluted soils.

## 1. Introduction

Soils are a natural source of heavy metals (HM), but geologic and anthropogenic activities have increased concentration thereof in soil, water and living systems [1]. Upon rapid and huge HM accumulation, remediation has become necessary to protect the environment from the toxic effects of HM. Several efforts have been thus made so far, to develop sustainable and environment-friendly strategies.

As an adjunct to the various phytoremediation approaches, the possibility of using soil bacteria together with plants has been increasingly explored [2]. These bacteria comprise bio-degradative bacteria, plant growth-promoting bacteria (PGPB) and bacteria that facilitate phytoremediation by other means [3].

PGPB enhance the growth of plants through various plant growth-promoting (PGP) traits (e.g., phosphate solubilization, production of indole-3-acetic acid, siderophores, ammonia, hydrogen cyanide and nitrogen fixation) [3,4,5,6]. In addition, they have the potential for metal detoxification and mitigation of plant’s stress in polluted environment.

Among HM, arsenic needs special attention. This element is included in the list of agents, prepared by the International Agency for Research on Cancer (IARC), for which sufficient evidence of carcinogenicity in humans exists (Group 1; [7]), and is ranked first in the 2017 CERCLA Priority List of the US Agency for Toxic Substances and Disease Registry [8]. For this reason, arsenic contamination of soil and groundwater, originated from natural and anthropogenic sources, can determine a price decline in contaminated agricultural food products. In addition, long-term exposure to arsenic from drinking arsenic-rich water is a great threat to public health [9].

Arsenic and its more than 200 compounds are ubiquitous in the environment and can be classified into three major groups: inorganic arsenic compounds, organic arsenic compounds, and arsine gas [7]. Toxicity depends on the form: soluble inorganic species are more toxic than organic forms, and arsenite (arsenic (III)) is more toxic than arsenate (arsenic (V); [10]). The trivalent and pentavalent forms are the most common oxidation states.

Pathways involved in resistance to both arsenite and arsenate are widely found in different species of Gram-negative and Gram-positive bacteria. Usually, As(V) is reduced to As(III) by an intracellular thiol-linked reductase, named ArsC [11], and arsenite is extruded from the cell using efflux proteins (ArsB or Acr3) that can act as proton exchanger, working alone or in conjunction with an intracellular ATPase (ArsA; [12]).

Plants subjected to arsenic have been shown to increase their susceptibility to pathogen infections [13]. In this framework, the combination of biotic and abiotic stress was responsible for a reduction in plant growth and yield. However, when rice plants were treated with selected *Pantoea* strains, such as EA106, a lower susceptibility to pathogens was observed as well as a lower arsenic uptake, giving the first evidence of an up-regulation of defense-related genes mediated by *Pantoea* [13]. In parallel, it has been demonstrated that the arsenic resistance gene cluster that confers arsenic resistance through arsenate reduction and arsenite efflux (*arsRBC* or *arsRBCH*) occurs in the genome of several strains belonging to different *Pantoea* species [14]. Based on these findings, it is important to understand the impact that arsenic tolerant *Pantoea* strains can have on reducing plant stress and improving plant growth performance.

*Pantoea agglomerans* is a representative member of *Pantoea* genus, and comparative genomics allows including *P. agglomerans* strains into different clades which comprise both clinical and plant-beneficial strains [15]. Some strains are frequently found in association with plant hosts [16], and others are agronomically relevant for their PGP features, exerted also by synthesis of indole-3-acetic acid (IAA), for the biocontrol activity and for their involvement in plant disease management [17]. Sequencing and annotation of the complete genome of plant-beneficial *P. agglomerans* strains can hence improve understanding of the potential use of these microorganisms as plant biostimulants and can be valuable for the identification of novel plant growth-promoting rhizobacteria (PGPR) to use to maximize the remediation potential of plants.

In the present study, the complete genome sequence of a strain of *P. agglomerans* (C1 strain) and a comparative genome analysis, specifically focused on PGP traits and heavy metal resistance, are reported. The strain was previously isolated from the phyllosphere of lettuce (*Lactuca sativa* L.) plants treated with vegetal-derived protein hydrolysates [18]. It showed biocontrol activity against *Erwinia amylovora*, as well as PGP traits, such as production of auxin-like phytohormones, after being placed in the genus *Pantoea* by 16S rDNA sequencing [19]. Results about experimental inoculation of plants and cultivation in the presence of arsenate and arsenite are also presented, so as to get insights into the potential of the *P. agglomerans* strain C1 to both survive in arsenate-contaminated soil and stimulate plant growth.

## 2. Materials and Methods

### 2.1. DNA Extraction, Genome Sequencing, Assembly and Annotation

Genomic DNA was extracted by using PureLink Genomic DNA Mini Kit and quantified by Qubit ds HS assay kit (Thermo Fisher Scientific Italia, Rodano (MI), Italy), as reported elsewhere [19]. Library preparation and genome sequencing were performed at Bio-Fab Research s.r.l. (Rome, Italy) using the Illumina MiSeq version 3 sequencing platform system in 300-nucleotide (nt) paired-end mode, and run statistics were determined using CLC Genomics Workbench 12 (Qiagen GmbH, Hilden, Germany). The Illumina generated reads were assembled by the A5-myseq assembly pipeline [20], as described in Luziatelli et al. [21]. Gene prediction analysis and functional annotation of the genome were performed by Rapid Annotation by using Subsystems Technology (RAST; [22]), specifically by the RAST Toolkit (RASTtk) option [23] and visualized with the SEED viewer [24].

### 2.2. Phylogenetic Tree Construction and ANI

The phylogenetic tree was constructed from user-selected genomes by the FastTree method [25] using the Phylogenetic Tree Building Service available at the Patric website (https://www.patricbrc.org), with all shared proteins as option and 1000 bootstrap replications. The analysis included: type-strains of major *Pantoea* species; five strain for each of the two *P. agglomerans* group; representative strains of *P. ananatis* and *P. vagans*; *Pantoea* strains that share a highly similar heavy metal resistance gene cluster; *Providencia rettgeri* DSM1131, as outgroup. Average nucleotide identity analysis was performed between *P. agglomerans* C1 and other *Pantoea* isolates included in the phylogenetic tree, using an on-line ANI calculator [26], and the presence of plasmid replicons or prophages was determined using the PlasmidFinder tool [27] and PHAge Search Tool Enhanced Release (PHASTER) [28,29], respectively.

### 2.3. Functional Genome Annotation and Identification of Genomic Islands

The Cluster of Orthologous Groups (COG) functional categories were assigned through the WebMGA server [30]. Homologs of genes contributing to plant growth promotion were identified with tBLASTn, using target protein sequences from closely related species, and functional genes involved in heavy metal resistance genes were identified by the bidirectional best hit analysis performed in RAST. Putative genomic islands (GI) generated from HGT were detected using IslandViewer 4 [31].

### 2.4. Production of Indole-3-Acetic Acid

To induce production of indole-3-acetic acid (IAA), 1 mL (~10^9^ cells) of an LB-medium overnight culture of strain C1 was transferred into a 100 mL Erlenmeyer flask containing 20 mL of LB medium supplemented with sterile-filtered tryptophan (0.4 mM). The liquid culture was grown at 30 °C in agitation (180 rpm), and cells were separated from the exhausted medium by centrifugation (10,000× *g* for 10 min) and discarded after 24 h. The collected supernatant was filtered through a 0.22 µm membrane and stored at −20 °C for later use. Total IAA was determined by a colorimetric method using Salkowski reagent and authentic IAA (Sigma-Aldrich, St. Louis, MO, USA) as a standard [32].

### 2.5. Determination of Siderophore Production

The siderophore production was detected by the Chrome Azurol-S assay [33], cultivating the microorganism on solid (Chrome Azurol agar, CAS; Merck KGaA, Darmstadt, Germany) or liquid medium (LB with 0.4 mM tryptophan or Fe-deficient King’s B medium). On agar plates, production was visualized as an orange halo around the colonies after 48 h incubation at 30 °C and was expressed according to the formula
(1)*W*_act_ = (*S*_h_^2^)/(*S*_c_ * *t*)

by Hrynkiewicz et al. [34], where *W*_act_ is the coefficients of activity, *S*_h_ is the diameter of the hydrolysis zone, *S*_c_ is the colony diameter, and *t* is the incubation time.

For quantitative analysis, 0.5 mL of an LB overnight culture of *P. agglomerans* strain C1 was transferred to a 250 mL Erlenmeyer flask containing 50 mL of the test medium. Cultures were grown at 30 °C and 180 rpm agitation speed. After 48 h, cultures were centrifuged at 10,000 rpm for 10 min and the resulting supernatant was filtered through a 0.22 μm pore size.

For siderophore quantification 0.5 mL of filtered supernatant were mixed with 0.5 mL of CAS assay solution, prepared as described by Alexander and Zuberer [35]. After reaching the equilibrium (20 min of incubation) the absorbance was measured spectrophotometrically at 630 nm using a reference containing 0.5 mL CAS solution with 0.5 mL uninoculated medium.

Siderophore production is expressed as percentage of siderophore units (PSU), calculated using the following formula:
(2)
[(*A*r − *A*s)/*A*r] * 100,

where *A*r is the Absorbance of reference (CAS assay solution + uninoculated media) and *A*s is the Absorbance of the sample (CAS assay solution + cell-free supernatant).

The experiment was performed in triplicate. In order to avoid iron contamination (on iron-deficient-cultures), all glassware was soaked in 10% nitric acid, overnight, and, subsequently, washed with deionized water prior to use.

### 2.6. Determination of Minimal Inhibitory Concentration of Arsenic

Minimal inhibitory concentration (MIC) of arsenite (As(III)) and arsenate (As(V)) for *P. agglomerans* strain C1 was determined in 20 mL cultures grown in 100 mL Erlenmeyer flasks at 30 °C in agitation (180 rpm). Cultivation was carried out in LB (Lennox) broth amended with sodium arsenite (As(III)) or sodium arsenate (As(V)) at a concentration between 5 and 50 mM. The stock solutions (200 mM) of sodium arsenate (Na_2_HAsO_4_·7H_2_0; Merck KGaA, Darmstadt, Germany) or sodium arsenite (NaAsO_2_ Merck KGaA, Darmstadt, Germany) were prepared in sterile water. Cultures were inoculated with LB overnight cultures (initial OD_600_ of 0.1), and growth was determined by OD measurement 48 h after the inoculum. All samples were tested in triplicates and medium without inoculation or medium inoculated with *Escherichia coli* strain JM109 were used as controls.

### 2.7. Plant Inoculation

Tomato cuttings experiments were carried out as reported previously by Colla et al. [36]. In brief, tomato seeds (*Solanum lycopersicum* L. cv. Marmande, SAIS Sementi, Cesena, Italy) were sown in moist vermiculite:peat-based substrate (1:1 volume ratio) in a germination tray, and incubated in a growth chamber. The growth chamber was set up to maintain a 16 h photoperiod with 25 °C light/18 °C night and 65% relative humidity. The average photosynthetic photon flux at the canopy level was 75 μmol m^−2^ s^−1^. After two weeks, the tomato seedlings, at a three true leaves stage, were cut at the base of the stem, and the obtained cuttings were dipped for 5 minutes into sterilized distilled water or sterilized distilled water supplemented with fresh LB medium (15 mL L^−1^); overnight culture (spent medium with cells; 15 mL L^−1^); filtered supernatant (cell-free spent medium; 15 mL L^−1^); indole-3-butyric acid solution (IBA; 500 mg L^−1^). IBA was dissolved in NaOH (1 M) and diluted in water to a final stock concentration of 1 g L^−1^. After treatment, seedlings were transplanted directly into plastic pots containing 8 cm of wetted perlite, as rooting medium, and, 15 days after planting, tomato cuttings were separated into shoots and roots. Roots were kindly washed with distilled water, to remove any perlite particles, and determination of root surface was done by using WinRHIZO Pro (Regent Instruments Inc., Quebec, Canada), connected to a STD4800 scanner. Ten cuttings were used for each treatment, and results were the mean value of three replicates for each treatment (with a total of 30 plants per treatment).

### 2.8. Statistical Analysis

Differences between treatment groups were compared using One-way analysis of variance (ANOVA) test, followed by Tukey’s honestly significant difference (HSD) test with significance set at *p* < 0.05.

### 2.9. Nucleotide Sequence Accession Number

The genome sequence of *P. agglomerans* C1 is available under NCBI BioProject PRJNA523737, with GenBank accession number SMLN00000000.1 and Sequence Read Archive (SRA) accession number SRP212904.

Accession numbers of the genomes used for phylogenetic analysis are reported in Appendix A.

## 3. Results and Discussion

### 3.1. Genome Sequencing and Comparison with Pantoea Genomes

In order to investigate the genomic features associated with strain C1, the whole genome was sequenced using Illumina MiSeq (300-bp paired end) technology [21]. As reported before, the complete genome consisted of one circular chromosome of 4,846,925-bp, with a GC content of 55.2% [21]. Building the reference sequence, NCBI re-annotated the C1 genome using NCBI Prokaryotic Genome Annotation Pipeline (GeneBank reference: SMLN00000000.1), which allowed us to re-estimate the number of genes, coding sequences, rRNAs and ncRNAs present in this genome (Table 1).

In agreement with data from agarose gel electrophoresis analysis of total genomic DNA, no plasmid was detected by using PlasmidFinder [27]. In contrast, a computer search by PHASTER [28,29] revealed the presence of two distinct large intact prophage regions exhibiting similarity with phages from *Erwinia amylovora* (ENT90; GenBank No. NC_019932) and *Salmonella enterica* serovar Enteritidis LK5 (RE_2010; GenBank No. HM770079) [21]_._ It is interesting to highlight that selected *P. agglomerans* strains (i.e., EH21-5) can be successfully utilized to develop effective phage therapies against plant pathogens, such as *E. amylovora* [37].

Whole-genome phylogenetic analysis revealed that strain C1 clustered in the same clade which includes *P. agglomerans* type strain DSM3493 (Figure 1).

The relationships obtained using the distance approach, based on the Average Nucleotide Identity (ANI), were congruent with the species tree showed in Figure 1; the similarity between genomes of strain C1 and *P. agglomerans* type strain DSM3493 and other *P. agglomerans* strains was about 99% (Table 2).

Based on these data, the strain can be reclassified as *P. agglomerans* C1. All the data also indicated that strain ZBG6 should belong to the species *P. agglomerans* rather than *P. vagans*, as formerly proposed (Figure 1).

Exploitation of strain C1 genome with IslandViewer 4 revealed the presence of 29 putative genomic islands (Appendix A), eleven of which had a size higher than 20,000 bp and whose reliability was supported by three different computational methods. Interestingly, a total of 9 out of the aforesaid 11 GI harbor phage- or mobile-related coding sequence (Appendix A).

### 3.2. Plant Beneficial Properties of Pantoea agglomerans C1

The protein-encoding genes (PEGs) predicted using RASTtk were classified into 18 functional categories based on COG of proteins [38]. As shown in Table 3, most of the genes were associated with functions, such as transcription (K; 8.71%), amino acid transport and metabolism (E; 8.33%), inorganic ion transport and metabolism (P; 6.11%), carbohydrate transport and metabolism (G; 5.88%), and cell wall/membrane/envelop biogenesis (M; 5.43%).

Nearly one-third of the entire set of genes encoding proteins cannot be annotated with a known function (Table 3).

Functional analysis of *P. agglomerans* C1 genome showed the presence of several genes contributing directly or indirectly to PGP and biocontrol activities (Table 4).

We identified the genes encoding key enzymes involved in the synthesis and secretion of IAA through the IPyA (*ipdC*) and the IAM (*amiE*) pathways [39]. In *P. agglomerans* C1 genome, we also found two operons (*speAB* and *speDE*) that could be involved in spermidine biosynthesis, a class of compounds that are essential for eukaryotic cells viability and have been correlated with lateral root development, pathogen resistance, and alleviation of oxidative, osmotic and acidic stresses [40]. The annotation study also revealed the presence of several gene clusters involved in mineral phosphate solubilization, including the genes encoding PQQ-dependent glucose dehydrogenase (*gcd*), membrane-bound gluconate-2-dehydrogenase (*gad*) and phosphatase-specific transport system (Table 3) [41,42].

As regards the indirect means of plant growth promotion, in *P. agglomerans* C1 genome, we found (Table 4) genes encoding enzymes involved in the synthesis of volatile organic compounds (acetoin and 2,3-butanediol; [43,44], Gamma-Aminobutyric Acid (GABA) [45], and siderophores [46], as well as genes encoding the three components of EfeUOB transporter, a ferrous iron transporter induced by low pH and low iron [47].

### 3.3. Effects of Pantoea Agglomerans C1 Cells and Metabolites on Root Growth

Strain C1 produced siderophores in both solid and liquid medium. Production on CAS agar medium was visualized in an orange halo around the colony, with a coefficient of activity (*W*_act_) of 0.21 ± 0.1. The highest siderophores production in liquid medium was obtained on King’s B after 48 h incubation (11 ± 0.5 PSU). However, the production of IAA was very limited in this medium, even in the presence of tryptophan (about 20 ± 1 mg of IAA for liter). In contrast, strain C1 produced IAA up to 150 ± 5 mg/L and siderophores up to 4.5 ± 0.5 PSU in LB supplemented with tryptophan (4 mM). For this reason, all experiments with tomato plants were carried out using cells and secreted metabolites from cultures grown in LB medium with tryptophan. Treatment of tomato shoots with the spent medium containing cells and secreted metabolites enabled a significant increase in root surface area, 2-weeks after application, with respect to the control shoots treated with distilled water (Figure 2, panel A).

This effect was comparable to that obtained in IBA-treated shoots (Figure 2, panel A). In contrast, treatment of tomato shoots with fresh LB medium had no effect on root growth compared to control shoots, thus indicating that this stimulatory effect was not dependent upon LB medium components.

When the cell-free supernatant collected from these cultures was used, the increase in root growth was even more remarkable (2-folds compared to water control and 1.45-fold compared to commercial IBA; Figure 2, panel A). The overall effect was found to be dose-dependent and, at higher doses, the increase in root surface was less pronounced (Figure 2, panel B).

Visual inspection of seedlings also indicated that application of strain C1 extracellular metabolites determined an increase in the number and length of major roots of tomato cuttings (Figure 2, panel C). These in vivo experiments clearly demonstrate that strain C1 produces metabolites that promote plant growth.

These results allow inferring that metabolites produced in vitro by strain C1 efficiently act as biostimulants. Although the biotechnological use of beneficial *Pantoea* strains is generally hampered by biosafety concerns, arising from clinical evidences that some strains are opportunistic human pathogens, and discrimination between clinical and plant beneficial strains cannot be achieved by phylogenetic analysis [17,48], this study shows that it can be taken advantage of the plant growth-promoting properties of the strain C1. A direct inoculation of the plant with bacterial cells and the release of the strain in the environment can be avoided. The results expand the range of potential applications of strain C1 and allow the development of novel biostimulants with low environmental impact, as well as the avoidance of the known problems related to competition between bioinoculants and soil-plant microbiome.

In vitro assays, performed in collaboration with IRBM Scientific Park (Pomezia, Italy), for testing cytotoxic activity of cell-free supernatant obtained from C1 cultures showed no anti-proliferative effect on HeLa cells, providing preliminary evidence of the biosafety of strain C1 extracellular metabolites.

### 3.4. Tolerance to Heavy Metals in Pantoea Agglomerans C1

In *P. agglomeran*s C1 genome we also found, distributed on different contigs, a number of genes related to resistance to toxic metals, including arsenic, copper and cadmium (Table 5).

For arsenical resistance, we identified three different gene clusters: *arsRH, arsRBC*, and *arsR-acr3*. The first cluster (*arsRH*) contains genes encoding a putative repressor (ArsR) and an NAD(P)H-dependent FMN reductase (ArsH) involved in the oxidation of arsenite to arsenate [49]. The *arsRBC* operon encodes a trans-acting transcriptional repressor protein (ArsR), belonging to the SmtB/ArsR family of metalloregulatory proteins, a putative arsenite antiporter (ArsB), and an arsenate reductase (ArsC) that reduces arsenate to arsenite [50]. The last operon (*arsR-acr3*), which is located in pro-phage_2, encodes for a putative transcription factor (ArsR), belonging to the metalloregulator SmtB/ArsR family, and an arsenite efflux pump (Acr3), belonging to ACR3 family [51].

Independent *cue* (copper efflux), *cus* (copper sensing) and *pco* (copper resistance) systems and accessory genes, which confer copper tolerance in bacteria, were also present (Table 5). The *cueR-copA* gene cluster encodes a putative copper-exporting P-type ATPase (CopA) and a two-component signal transduction system (CusR/CusS), involved in maintaining metal ion homeostasis, which activates, under anaerobic conditions, the expression of the *cus*CFBA operon in response to elevate concentration of copper [52]. The last set of genes includes homologues to the copper-inducible *copABCD* and *pcoRS* gene cluster encoding a two-component regulatory system (PcoR/PcoS) and four structural proteins including an inner membrane protein (CopD), an outer membrane protein (CopB) and two periplasmic proteins (CopA, CopC; [53]. CopA is a multi-copper oxidase protein, responsible for the oxidation of Cu(I) in the periplasmic space, which confers high resistance to copper [54].

We also identified two genes (*czcA* and *czcC*) encoding a putative cadmium resistance protein (CzcA) and an RND efflux outer membrane protein (CzcC), respectively (Table 5). These genes belong to the *czc* efflux system and are involved in Cu/Zn/Co detoxification in many bacteria [55].

Interestingly, most of the genes involved in tolerance against heavy metals are clustered in a 23.9-Kb region on contig 2 (endpoints: 346889–370816; Table 5) and are included in one of the GI supported by all computational methods of IslandViewer 4 (Appendix A). A genome-mining analysis showed that this gene cluster from *P. agglomerans* C1 existed in eight *Pantoea* strains belonging to different species (*P. eucrina*, *P. ananatis* and *P. agglomerans*; Figure 1). Regardless of the absolute genetic distance among the genomes, the structure of the heavy metal resistance gene (MRG) cluster was conserved, and the overall nucleotide sequence identity of the 23.9-Kb region ranged from 96% to 98% (Figure 3).

This high degree of sequence identity and their location on a GI (at least on strain C1) suggests that the acquisition of these genes can occur upon horizontal gene transfer (HGT) events.

In order to determine potential selective advantage, due to the three *ars* gene clusters, we evaluated the maximum tolerable concentration (MTC) of *P. agglomerans* C1 for arsenate and arsenite. Data reported in Figure 4 (panel A) indicate that strain C1 was able to grow in medium amended with arsenate (As(V)) up to 100 mM, while *E. coli* control strain grew up to 20 mM.

In contrast, no difference was observed for As(III) MTC; both strains grew in medium containing arsenite up to 1 mM (Figure 4, panel B). Our findings confirmed that the *ars* genes confer a competitive advantage to C1 cells growing in the presence of As(V) and indicated that the minimal inhibitory concentration (MIC) of this strain for arsenate was similar to that reported for *P. agglomerans* IMH [56] and arsenate-reducing bacteria isolated from arsenic-contaminated sites [57].

## 4. Conclusions

In conclusion, we demonstrated that metabolites produced by *P. agglomerans* C1 elicit promotion of plant growth, and the complete genome provides useful insights into the mechanisms underlying the PGP-traits. Importantly, the functional analysis of *P. agglomerans* C1 genome suggested that this strain has the potential to survive and grow in environments contaminated by heavy metals and can be used as a plant growth-promoting bacterium in heavy metal polluted soils. Finally, we provided evidence that strain C1 probably acquired the genes related to resistance to toxic metals by horizontal gene transfer.

Furthermore, the identification of several genes contributing in plant growth-promotion (i.e., lateral root development, pathogen resistance, and alleviation of oxidative, osmotic and acidic stresses) and conferring plant resistance to heavy metals (i.e., arsenic, copper, cadmium), strengthens the use of *P. agglomerans* C1 in reduction of biotic and abiotic stress response in heavy metals polluted soil, by improving plant growth performance.

## Figures and Tables

**Figure 1 microorganisms-08-00153-f001:**
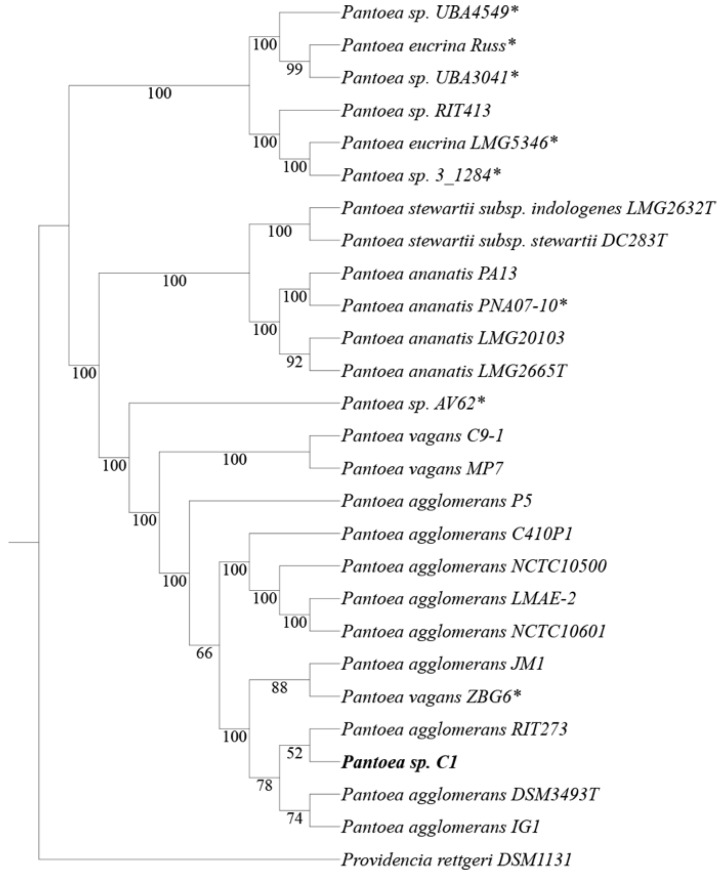
Comparison analysis of strain C1 with other 25 *Pantoea* strains. The phylogenetic tree was built from user-selected genomes by the FastTree method [25]. Branch labels represent bootstrap support (in percent; 1000 bootstrap replicates). T indicates type strain; the asterisk (*) indicates *Pantoea* strains that share a highly similar heavy metal resistance gene cluster (see Figure 3). *Providencia rettgeri* DSM1131 was used as outgroup. Accession numbers are reported in Appendix A.

**Figure 2 microorganisms-08-00153-f002:**
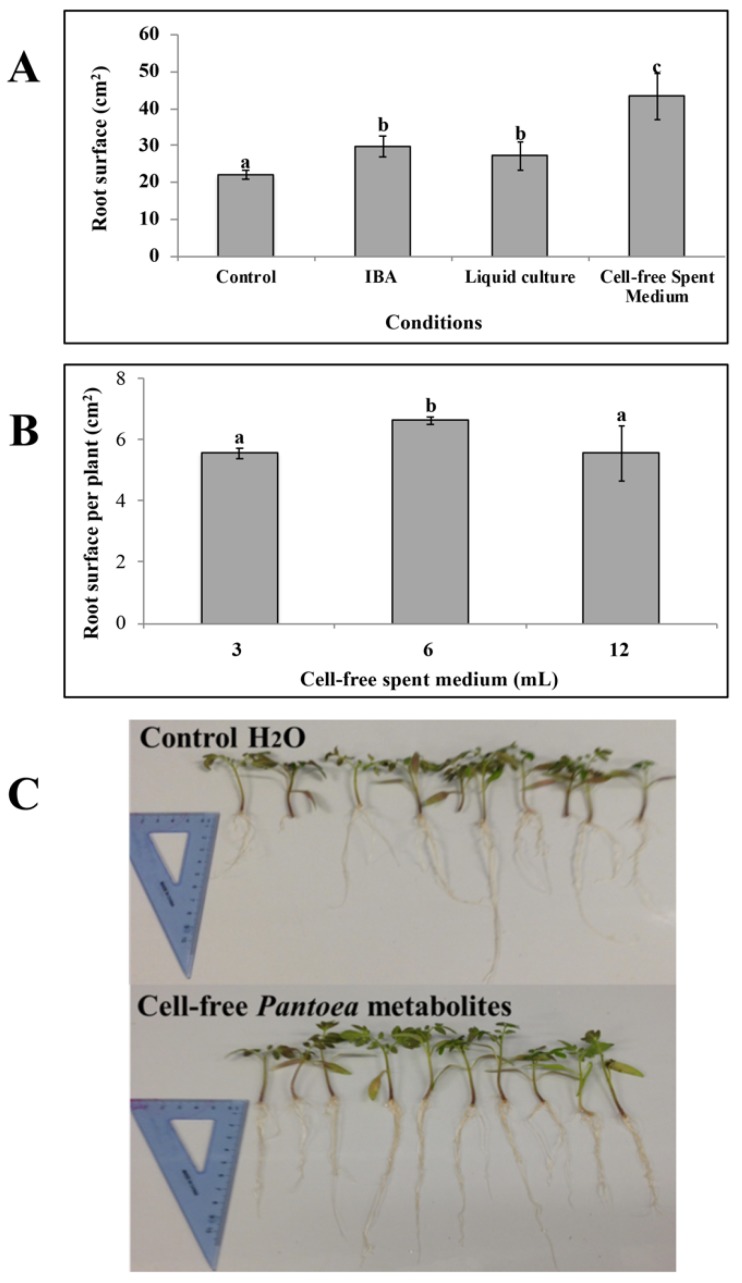
C Effect of *P. agglomerans* C1 and its metabolites on the root characteristics of tomato (*Solanum lycopersicum* L.) cuttings. (**A**): Effect of C1 culture (spent medium with cells), C1 metabolites (cell-free spent medium) and indole-3-butyric acid solution (IBA) application on total root surface of tomato cuttings 2-weeks after treatment. (**B**): Dose response showing the effect of C1 metabolites application on total root surface of tomato cuttings 2-weeks after treatment with 3, 6 or 12 mL/L of cell-free *P. agglomerans* C1 metabolites. (**C**): Differences of root abundance and appearance of tomato cuttings 2-weeks after immersion in a solution containing 0 (Control) or 6 mL/L of cell-free *P. agglomerans* C1 metabolites. The spent medium with cells and the cell-free spent medium contained an IAA concentration of 105 ± 10 μg mL^−1^. Each data point is the mean ± SE of 10 replicates. Values with no letter in common significantly differ at *p* < 0.05 (Tukey HSD test).

**Figure 3 microorganisms-08-00153-f003:**
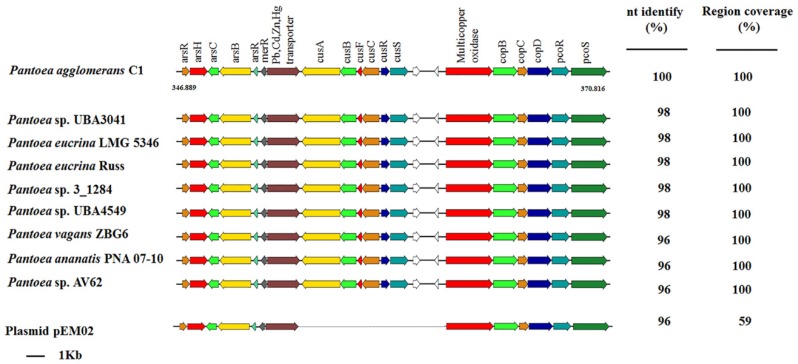
Organization of the heavy metal gene (MRG) cluster of *P. agglomerans* C1 and comparison with other *Pantoea* genomes that have the same 19-gene cluster and with pEM02 plasmid from *Erwinia* sp. EM595 (GenBank reference: LN907829.1). Genes with unknown function are indicated in white. Genome accession numbers are reported in Appendix A.

**Figure 4 microorganisms-08-00153-f004:**
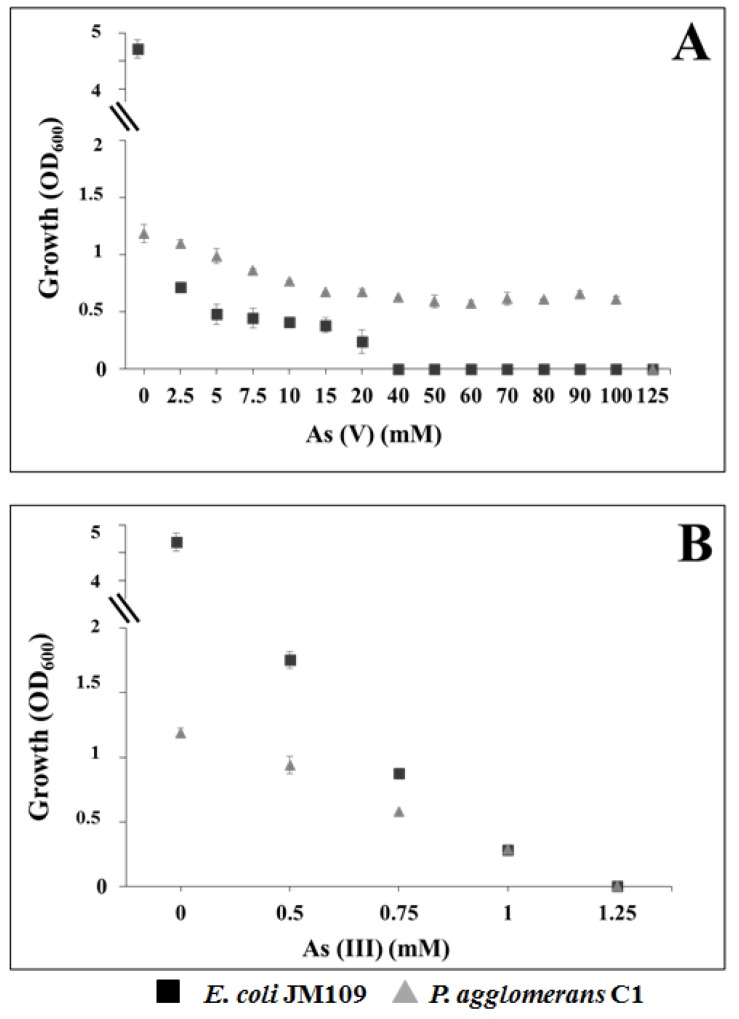
Growth of *P. agglomerans* C1 and *E. coli* K12 derivative JM109 in LB medium supplemented with increasing concentrations of arsenate, As(V) (**A**), and arsenite, As(III) (**B**).

**Table 1 microorganisms-08-00153-t001:** General features of *P. agglomerans* C1 genome.

Species	*Pantoea agglomerans*	Source
Strain	C1	
Assembly level	Contig	
No. of sequences	22	[21]
Genome size (bp)	4,846,925	[21]
GC content (%)	55.2	[21]
Gene	4601	This work
*CDS*	4497	This work
*RNA*	104	This work
rRNA (5S,16S,23S)	9, 6, 9	This work
Completed (5S,16S,23S)	9, 1, 1	This work
Truncated (16S,23S)	5, 8	This work
tRNA ncRNA	70 10	[21] This work
Prophage	2	This work
Genomic island (integrated method)	11 > 20,000 bp	This work

**Table 2 microorganisms-08-00153-t002:** Average Nucleotide Identity (ANI) values (in percentages) based on alignment of the whole genome of strain C1 and the most closely related members of the genus *Pantoea*.

CODE	STRAIN	1	2	3	4	5	6	7	8	9	10	11	12	13	14	15
1	Strain C1	*	98.7	98.7	98.7	98.6	97.8	97.3	84.0	84.1	84.2	84.3	84.1	91.3	91.3	98.7
2	*Pantoea agglomerans* RIT273	98.7	*	98.7	98.7	98.7	97.9	97.3	84.0	84.0	84.0	84.1	84.1	91.3	91.3	98.8
3	*Pantoea agglomerans* DSM3463^T^	98.7	98.7	*	98.7	98.7	97.9	97.3	84.1	84.0	84.0	83.9	84.1	91.3	91.4	98.7
4	*Pantoea agglomerans* JM1	98.7	98.7	98.7	*	98.7	97.8	97.3	84.0	84.0	84.0	83.9	84.1	91.3	91.3	98.8
5	*Pantoea agglomerans* IG1	98.6	98.7	98.7	98.7	*	97.8	97.2	83.9	83.9	83.9	83.9	84.0	91.3	91.3	98.7
6	*Pantoea agglomerans* C410P1	97.8	97.9	97.9	97.8	97.8	*	97.5	84.0	84.1	83.9	83.9	84.1	91.8	91.8	97.9
7	*Pantoea agglomerans* P5	97.3	97.3	97.3	97.3	97.2	97.5	*	83.9	84.0	84.0	83.9	84.0	91.2	91.3	97.3
8	*Pantoea ananatis* LMG2665^T^	84.0	83.9	84.1	84.0	83.9	84.0	83.9	*	99.3	99.2	83.9	86.0	84.1	84.2	84.0
9	*Pantoea ananatis* LMG20103	84.0	84.0	84.0	84.0	83.9	84.1	84.0	99.3	*	99.2	83.8	85.9	84.2	84.2	83.9
10	*Pantoea ananatis* PNA 07-10	84.2	84.0	84.0	84.0	83.9	83.9	84.0	99.2	99.2	*	84.2	85.9	84.2	84.0	84.2
11	*Pantoea eucrina* LMG5346^T^	84.3	84.1	84.0	83.9	83.9	83.9	83.9	83.9	83.8	84.2	*	84.0	83.8	83.8	84.2
12	*Pantoea stewartii* sub. *stewartii* DC283^T^	84.1	84.1	84.1	84.1	84.0	84.1	84.0	86.0	85.9	85.9	84.0	*	84.2	84.1	84.0
13	*Pantoea vagans* C9-1	91.3	91.3	91.3	91.3	91.3	91.8	91.2	84.1	84.2	84.2	83.8	84.2	*	96.9	91.3
14	*Pantoea vagans* MP7	91.3	91.3	91.4	91.3	91.3	91.8	91.3	84.1	84.2	84.0	83.8	84.1	96.9	*	91.3
15	*Pantoea vagans* ZBG6	98.7	98.8	98.7	98.8	98.7	97.9	97.3	84.0	83.9	84.3	84.2	84.0	91.3	91.3	*

^T^ Type-strain. * = 100.

**Table 3 microorganisms-08-00153-t003:** Number of genes associated with general Clusters of Orthologous Groups (COG) functional categories.

Function	Code	Value	%age	Description
CELLULAR PROCESSES AND SIGNALING	D	62	1.32	Cell cycle control, cell division, chromosome partitioning
	M	255	5.43	Cell wall/membrane/envelope biogenesis
	N	95	2.02	Cell motility
	O	107	2.28	Post-translational modification, protein turnover, and chaperones
	T	106	2.26	Signal transduction mechanisms
	U	54	1.15	Intracellular trafficking, secretion, and vesicular transport
	V	47	1.00	Defense mechanisms
INFORMATION STORAGE AND PROCESSING	A	0	0.00	RNA processing and modification
	B	0	0.00	Chromatin structure and dynamics
	J	193	4.11	Translation, ribosomal structure and biogenesis
	K	409	8.71	Transcription
	L	158	3.36	Replication, recombination and repair
METABOLISM	C	234	4.98	Energy production and conversion
	E	391	8.33	Amino acid transport and metabolism
	F	106	2.26	Nucleotide transport and metabolism
	G	276	5.88	Carbohydrate transport and metabolism
	H	176	3.75	Coenzyme transport and metabolism
	I	113	2.41	Lipid transport and metabolism
	P	287	6.11	Inorganic ion transport and metabolism
	Q	39	0.83	Secondary metabolites biosynthesis, transport, and catabolism
POORLY CHARACTERIZED	R	0	0.00	General function prediction only
	S	937	19.95	Function unknown
	-	651	13.86	Not in COGs

**Table 4 microorganisms-08-00153-t004:** Genes potentially associated with Plant Growth-Promotion traits in *P. agglomerans* C1.

**Direct Plant Growth-Promotion**			
**Gene**	**EC No.**	**Annotation**	**Gene Location, Coding Strand (+/−)**
**IAA production**			
*ipdC*	4.1.1.74	Indole-3-pyruvate decarboxylase	Contig1: 2029913-2028261, −
*amiE*	3.5.1.4	Aliphatic amidase	Contig1: 254208-254999, +
*aec*		Auxin efflux carrier family protein	Contig1: 1779607-1780566, +
**Spermidine biosynthesis**			
*speA*	3.5.3.11	Agmatinase	Contig4: 165937-1659017, −
*speB*	4.1.1.19	Biosynthetic arginine decarboxylase	Contig4: 168083-1686107, −
*speD*	4.1.1.50	S-adenosylmethionine decarboxylase proenzyme	Contig3: 73489-74298, +
*speE*	2.5.1.16	prokaryotic class 1A Spermidine synthase	Contig3: 73489-74298, +
**Phosphate solubilization**			
*gad*	1.1.99.3	Gluconate 2-dehydrogenase, membrane-bound, cytochrome c	Contig3: 303092-301830, −
*gad*	1.1.99.3	Gluconate 2-dehydrogenase, membrane-bound, flavoprotein	Contig3: 304866-303097, −
*gad*	1.1.99.3	Gluconate 2-dehydrogenase, membrane-bound, gamma subunit	Contig3: 305631-304903, −
*gad*	1.1.99.3	Gluconate 2-dehydrogenase, membrane-bound, cytochrome c	Contig3: 495485-494175, −
*gad*	1.1.99.3	Gluconate 2-dehydrogenase, membrane-bound, flavoprotein	Contig3: 497280-495496, −
*gad*	1.1.99.3	Gluconate 2-dehydrogenase, membrane-bound, gamma subunit	Contig3: 498017-497283, −
*gcd*	1.1.5.2	Glucose dehydrogenase pyrroloquinoline quinone (PQQ)-dependent	Contig3: 476263-478653, +
*pqq*		Coenzyme PQQ synthesis protein B,C,D,E,F	Contig1: 1076330-1081693, +
*phoU*		Phosphate transport system regulatory protein	Contig6: 207107-206373, −
*pstB*		Phosphate transport ATP-binding protein	Contig6: 207898-207125, −
*pstA*		Phosphate transport system permease protein	Contig6: 208833-207943, −
*pstC*		Phosphate transport system permease protein	Contig6: 209792-208830, −
*pstS*		Phosphate ABC transporter, periplasmic phosphate-binding protein	Contig6: 210923-209880, −
**Indirect Plant Growth-Promotion**			
**Gene**	**EC No.**	**Annotation**	**Gene Location, Coding Strand (+/−)**
**Volatile organic compounds (VOCs)**			
*alsR*		Transcriptional regulator of alpha-acetolactate operon	Contig7: 135886-136791, +
*alsD*	4.1.1.5	Alpha-acetolactate decarboxylase	Contig7: 135781-134999, −
*alsS*	2.2.1.6	Acetolactate synthase	Contig7: 134984-133305, −
*bdh*	1.1.1.41.1.1.304	2,3-butanediol dehydrogenase, S-alcohol forming, (R)-acetoin-specific/Acetoin (diacetyl) reductase	Contig7: 133283-132510, −
**GABA production**			
*gabD*	1.2.1.16	Succinate-semialdehyde dehydrogenase [NAD(P)^+^]	Contig4: 449240-447789, −
*gabT*	2.6.1.19	Gamma-aminobutyrate:alpha-ketoglutarate aminotransferase	Contig2: 419393-420679, +
**Siderophores biogenesis**			
*fes*		Enterobactin esterase	Contig3: 384712-385917, +
*entA*	1.3.1.28	2,3-dihydro-2,3-dihydroxybenzoate dehydrogenase	Contig3: 399161-399919, +
*entB*	3.3.2.1	Isochorismatase	Contig3: 398310-399164, +
*entC*	5.4.4.2	Isochorismate synthase	Contig3: 395486-396664, +
*entE*	2.7.7.58	2,3-dihydroxybenzoate-AMP ligase	Contig3: 396675-398291, +
*entF*	6.3.2.14	Enterobactin synthetase component F	Contig3: 386228-390157, +
*fepA*		TonB-dependent receptor; Outer membrane receptor for ferric enterobactin and colicins B, D	Contig3: 384461-382194, −
*fepB*		Ferric enterobactin-binding periplasmic protein	Contig3: 395308-394340, −
*fepC*		Ferric enterobactin transport ATP-binding protein	Contig3: 390992-390201, −
*fepD*		Ferric enterobactin transport system permease protein	Contig3: 392922-391966, −
*fepG*		Ferric enterobactin transport system permease protein	Contig3: 391969-390989, −
*entS*		Enterobactin exporter	Contig3: 393083-394345, −
*ybdZ*		Putative cytoplasmic protein YbdZ in enterobactin biosynthesis operon	Contig3: 386017-386235, +
*fhuA*		Ferric hydroxamate outer membrane receptor	Contig3: 51852-49651, −
*fhuC*		Ferric hydroxamate ABC transporter, ATP-binding protein	Contig3: 49611-48817, −
*fhuD*		Ferric hydroxamate ABC transporter, periplasmic substrate binding protein	Contig3: 48806-47928, −
*fhuB*		Ferric hydroxamate ABC transporter, permease component	Contig3: 47928-45949, −
**Ferrous iron transporter (EfeUOB)**			
*efeU*		Ferrous iron transport permease	Contig1: 1504038-1503214, −
*efeO*		Ferrous iron transport periplasmic protein contains peptidase-M75 domain and (frequently) cupredoxin-like domain	Contig1: 1504038-1503214, −
*efeB*		Ferrous iron transport peroxidase	Contig1: 1503155-1502046, −

**Table 5 microorganisms-08-00153-t005:** Genes for tolerance against heavy metal toxicity in *P. agglomerans* C1.

Gene	EC No.	Annotation	Gene Location, Coding Strand (+/−)
**Arsenic tolerance**			
***arsRH***			
*arsR*		Arsenical resistance operon repressor	Contig2: 346889-347179, +
*arsH*		Arsenic resistance protein ArsH	Contig2: 347176-347898, +
***arsRBC***			
*arsR*		Arsenical resistance operon repressor	Contig2: 350170-349817, −
*arsB*		Arsenic efflux pump protein	Contig2: 349720-348437, −
*arsC*	1.20.4.1	Arsenate reductase glutaredoxin-coupled, Glutaredoxin-like family	Contig2: 348387-347959, −
***arsR-acr3***			
*arsR*		Arsenical resistance operon repressor	Contig9: 2594-2229, −
*acr3*		Arsenical-resistance protein ACR3	Contig9: 2180-1200, −
**Copper tolerance**			
***cueR-copA***			
*cueR*		Transcriptional regulator, MerR family	Contig1: 198563-198967, +
*copA*	3.6.3.3 7.2.2.12 7.2.2.9	Lead, cadmium, zinc and mercury transporting ATPase Copper-translocating P-type ATPase	Contig1: 195952-198465, −
***cueR-copA***			
*cueR*		Transcriptional regulator, MerR family	Contig2: 350890-350432, −
*copA*	3.6.3.3 7.2.2.12 7.2.2.9	Lead, cadmium, zinc and mercury transporting ATPase Copper-translocating P-type ATPase	Contig2: 350970-353627, +
***copABCD_pcoRS***			
*copA*		Multicopper oxidase	Contig2: 364520-366361, +
*copB*		Copper resistance protein CopB	Contig2: 366396-367349, +
*copC*		Copper resistance protein CopC	Contig2: 367381-367761, +
*copD*		Copper resistance protein CopD	Contig2: 367766-368698, +
*pcoR*		Transcriptional regulator PcoR	Contig2: 368730-369410, +
*pcoS*	2.7.13.3	Sensory protein kinase PcoS	Contig2: 369407-370816, +
**Cadmium tolerance**			
***cusCFBA_cusSR***			
*cusC*		Cation efflux system protein CusC	Contig2: 360131-358746, −
*cusF*		Cation efflux system protein CusF	Contig2: 358717-358364, −
*cusB*		Cobalt/zinc/cadmium efflux RND transporter, Membrane fusion protein, CzcB family	Contig2: 358250-356958, -
*cusA*		Cation efflux system protein	Contig2: 356947-357380, −
*cusR*		Copper-sensing two-component system response regulator CusR	Contig2: 360326-361009, +
*cusS*		Copper sensory histidine kinase CusS	Contig2: 360999-362453, +
***czcAC***			
*czcA*		Cobalt-zinc-cadmium resistance protein CzcA	Contig2: 504650-501588, −
*czcC*		Probable Co/Zn/Cd efflux system membrane fusion protein	Contig2: 505738-504650, −

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
