# Peer review of "Genome Sequencing of Pantoea agglomerans C1 Provides Insights into Molecular and Genetic Mechanisms of Plant Growth-Promotion and Tolerance to Heavy Metals"

_microorganisms, 2020, doi:10.3390/microorganisms8020153_

Round 1

Reviewer 1 Report

Manuscript ID : microorganisms-663086

Genome sequencing of Pantoea agglomerans C1 provides insights into molecular and genetic mechanisms of plant growth-promotion and tolerance to heavy metals

Authors: Francesca Luziatelli, Anna Grazia Ficca, Mariateresa Cardarelli, Francesca Melini, Andrea Cavalieri, Maurizio Ruzz

The main concern is that the  manuscript contains data alredy published in : 

Luziatelli, F., Ficca, A.G., Melini, F., Ruzzi, M. Genome sequence of the Plant Growth-Promoting
404 Rhizobacterium (PGPR) Pantoea agglomerans C1. Microbiol. Resour. Announc. 2019, 8, e00828-19.

published by the same authors. All published data should be mentioned just by citing published work. Contrary it could be considered as auto-plagiarism.    

Author Response

All the revisions were highlighted in yellow in the revised manuscript.

Reviewer's comment: Auto-plagiarism.

Response: Following the Reviewer’s suggestion, published data have been mentioned indicating the appropriate reference. A special effort has been made to clarify the differences between the previously published data and novel data presented in this manuscript.

Reviewer 2 Report

This manuscript describes a functional analysis of a newly isolated Pantoea agglomerans strain. The authors specifically address the role of the bacterium in plant growth promotion. Overall, I found the paper to be a comprehensive analysis that incorporates a diversity of phenotypic testing and a clear connection to the genomics.

Specific Comments:
The introduction could be improved by connecting the two ideas, heavy metal resistance and plant growth promotion. As written, there are clearly two different goals with a sharp transition between the two ideas around L62. The conclusion section also suffers from this dual objective. The clarity of the study can be improved incorporating these two ideas and specifying the logic present in the conclusions.

There is significant overlap between the genome sequencing results and that previously published in the Genome Announcement. Other than the last two rows, Table 1 is previously published. It would be appropriate to indicate that the information has been previously analyzed. You could also more specifically state that you did not detect the prophages in your previous publication.

This could be a timing issue, but the genome sequence of the listed isolate is not included in the NCBI BioProject (PRJNA523737), but only the SRA raw data. This genome information should be made public before publication.

It is not clear why the results and discussion were combined. The two sections could be separated to increase the clarity of the manuscript.

How were the genomes shown in Figure 1 selected for analysis? Providing a more thorough explanation will provide needed transparency for the methods.

In Table 4, the first two gad annotations have the same genomic location. This should be corrected or explained.

In Figure 2 panel B, the labels on the x-axis need to be explained.

Table 5 seems to be organized by genome location, but Table 4 is sorted by Gene. Unless there is a specified reason, a single sorting method should be used for clarity. If this organization is related to the discussion of gene clusters, than that should be made more clear in the text.

The genome labels in Figure 3 are not standardized. In some cases the genus name is abbreviated. Including genbank accession numbers here would also help the reader.

Author Response

All the revisions were highlighted in yellow in the revised manuscript.

1. Reviewer's comment: "The introduction could be improved".

Response: Following Reviewer’s suggestions, the Introduction has been improved including new references that connect heavy metal resistance and plant growth promotion in Pantoea.

2. Reviewer's comment: "The conclusion can be improved".

Response: The conclusion has been implemented following Reviewer’s suggestions.

3. Reviewer's comment: "The overlap between the genome sequencing results and previously published data".

Response: Table 1 was revised, published data have been mentioned indicating the appropriate reference and an explanation of differences between published and new data have been included in the text (page 6, line 197-200).

4. Reviewer's comment: "Genome reference".

Response: The genome information has been made public and the reference was included in the main text.

5. Reviewer's comment: "Genomes selected for analysis".

Response: The strategy used for the selection of Pantoea genomes included in the phylogenetic analysis was detailed in Materials and Methods (page 3, line 109-112).

6. Reviewer's comment: "gad annotations in Table 4".

Response: gad annotations have been corrected.

7. Reviewer's comment: "Figure 2 panel B labels".

Response: Panel B and Figure 2 legend have been modified to enhance the presentation of the data.

8. Reviewer's comment: "Table 5 format".

Response: The table has been redrawn to conform to Table 4.

9. Reviewer's comment: "Figure 3 genome labels".

Response: The figure has been modified following Reviewer’s suggestions and a reference to the list of accession numbers for genomes utilized for phylogenetic analysis (Table S1) was included in the legend.

10. Reviewer's comment: "Results and Discussion".

Response: We hope that changes in the main text, tables and figures will be useful and sufficient to increase the clarity of the manuscript avoiding any separation of the Results and Discussion sections. In our opinion, bringing results on the annotation of genes and gene function and their discussion together, increases the clarity of the text.

Round 2

Reviewer 1 Report

Some comments

L 23. change arseniate to arsenate As (V)

L 92. change arseniate to arsenate

L355. change arseniate to arsenate

L359. change arseniate to arsenate

L311. Table5.  gene location is not described; add explication (+,- strands)

EC numbers are not uniforms (remove / )

annotations of genes are not aligned  to corresponding gene

Reviewer 2 Report

I appreciate the efforts of the authors to improve the clarity of the introduction. The rest of my previous comments have been addressed in the current revision.

Minor comments
L46, PGP is not defined.
Table 1, change Prophage Source to “This work”

Author Response

We would like to thank the Reviewer for his comments.

Line 46. "PGP traits" was changed to "plant growth-promoting (PGP) traits"

Table 1. Prophage Source was changed to “This work”